# A 6-Bit 20 GS/s Time-Interleaved Two-Step Flash ADC in 40 nm CMOS

Dong-Ryeol Oh

School of Electrical Engineering, Korea Advanced Institute of Science and Technology (KAIST), Daejeon 34141, Korea; droh@kaist.ac.kr; Tel.: +82-42-350-7525

**Abstract:** A 6-bit 20 GS/s 16-channel time-interleaved (TI) analog-to-digital converter (ADC) using a two-step flash ADC with a sample-and-hold (S/H) sharing technique and a gain-boosted voltage-to-time converter (VTC) is presented for high-speed wireline communication systems. By sharing one S/H between coarse and fine stages in the two-step flash ADC, the input bandwidth as well as area and power efficiency can be improved without a gain error between coarse and fine ADCs. Thanks to an eight-time interpolation using the gain-boosted VTC, the fine ADC has a small gate capacitance without a speed penalty, even in a small input voltage range. A prototype ADC implemented in a 40 nm CMOS process occupies a 0.1 mm$^2$ active area. The measured differential non-linearity (DNL) and integral non-linearity (INL) after offset and gain calibrations were 0.45 and 0.39 least significant bit (LSB), respectively. With a 9.042 GHz input, the measured signal-to-noise and distortion ratio (SNDR) and the spurious-free dynamic range (SFDR) were 30.12 and 40.23 dB, respectively. The small input capacitance of the sub-ADC enables a power-efficient track-and-hold amplifier (THA), resulting in a power consumption of 56.2 mW under a supply voltage of 0.9 V. The prototype ADC achieves a figure of merit (FoM) of 107.4 fJ/conversion-step at 20 GS/s.

**Keywords:** two-step; time-interleaving; time-domain; interpolation; voltage-to-time converter; sample-and-hold; flash; reference embedding; clock generation

## 1. Introduction

The demand for wired input/output bandwidth within data center networks is steadily increasing, driven by significant increases in data generation on wireline communication systems, such as cloud computing, mobile devices, and the Internet of things (IoT). In order to meet the demand, high-speed wireline communication systems applied with a multilevel signal modulation format, such as a pulse amplitude modulation 4-level (PAM-4), are required. In these wideband data communication systems, DSP-based high-speed serial links using an analog-to-digital converter (ADC) enable more complex and flexible applications of back-end digital signal processing. However, due to the addition of high-speed ADCs, the characteristics of input bandwidth, sampling rate, effective resolution, area, and power consumption of the ADC have a significant impact on the performance of the systems [1–7]. Recent studies show that a time-interleaving (TI) architecture is essential to convert the data above 10 GHz, and the performance of sub-ADC is a large portion of the TI ADC. At a medium resolution and a 20 GS/s conversion rate, sub-ADC types for the TI ADCs are generally divided into the flash and successive approximation register (SAR). Due to the fastest conversion speed of the flash ADC, the flash-based TI ADCs have the advantage of reducing the number of interleaving channels, which can reduce the hardware burden, such as a clock distribution and channel mismatch calibration on the TI ADC. However, many comparators required in the flash ADC and the burden on the offset calibration circuits for them directly affect the area and power consumption of the flash-based TI ADCs [6–10]. Recent studies on tens of GS/s TI ADCs have shown that the conversion speed of the SAR ADCs, used as the sub-ADC of the TI ADC, has been improved to the GHz level thanks to the advanced CMOS process [11–14]. However, the SAR

ADC requires not only the high-speed design for the comparator and logic corresponding to the number of conversion cycles, but also the management of internal clock signals for the comparator.

On the other hand, single-channel two-step flash ADCs have often been utilized to maximize the aforementioned advantages of the flash and SAR ADCs [15–21]. However, it was previously reported that the two-step flash ADCs have drawbacks, which are as follows: (1) settling time required to select the reference voltage ranges for the fine ADC (FADC) [15–17]; (2) large input capacitance and offset calibration complexity due to the full flash hardware in the FADC [18–20]; (3) bandwidth mismatch due to an additional input sampler [21]. Because of these limitations, SAR and flash architectures have been preferred over the two-step flash architecture as a sub-ADC for the TI ADC. With these reasons as a motivation, in this paper, a competitive TI two-step flash ADC suitable for a high-speed data conversion was proposed by improving the drawbacks of the existing two-step structures and applying them to the TI ADC.

The two-step flash ADC architecture used in this proposed TI ADC could be implemented with a low power, small area, and wide input bandwidth thanks to sample-and-hold (S/H) sharing and reference-embedding eight-time interpolation techniques [22]. The two-step flash ADC could guarantee inherent gain matching between the coarse and fine stage and could reduce the sampling capacitance as well by utilizing the S/H sharing technique. In addition, thanks to the reference-embedded interpolation technique, only one capacitive digital-to-analog converter (C-DAC) is required for the coarse stage, and a reference resistor-string (R-string) for the FADC and its settling speed burden are eliminated. As a result, the two-step flash ADC could be designed with a low-power and wide input bandwidth. The small input capacitance of the sub-ADC leads the input sampling network of the TI architecture, such as a track-and-hold amplifier (THA), to be fast and power efficient. Therefore, in this design, only four-channel THAs, each operating at 5 GHz, are used for the 20 GS/s sampling rate, which realize the wide input bandwidth and reduce the complexity of the channel mismatch calibration.

This paper is organized as follows. The overall ADC with an input network and sub-ADC is described in Section 2. Detailed circuit implementations of a pseudo-differential comparator for a coarse ADC (CADC) with an offset calibration capability, a gain-boosted voltage-to-time converter (VTC) for an FADC, and a high-speed clock generation scheme with a digitally-controlled delay line (DCDL) are explained in Section 3. Section 4 shows the measurement results and Section 5 concludes the paper.

## 2. Proposed ADC Architecture

A block diagram of the proposed 6-bit 20 GS/s TI two-step flash ADC is shown in Figure 1a. The TI ADC consists of 4-channel 5 GS/s input samplers with a source follower (SF) buffer (i.e., THA), 16-channel 6-bit 1.25 GS/s two-step flash ADCs with one S/H shared by 2.5b CADC and 4b FADC, multi-phase clock generators (CG) for 5 GHz main sampling clocks ($\Phi_{TH}$) and 1.25 GHz sub sampling clocks ($\Phi_S$ and $\Phi_{CM}$), memory for digital calibration, and a decimator for real-time measurement. Ten-gigahertz differential clocks ($\Phi_{BOP}$ and $\Phi_{BOM}$) are applied to the clock initial logic through the clock buffer based on the low-voltage differential signaling (LVDS) I/O [23]. As shown in Figure 1b, the clock initial logic ensures that the first rising edge of the positive output clock ($\Phi_{IOP}$) always precedes the first rising edge of the negative output clock ($\Phi_{IOM}$) whenever the ADC is enabled. In this way, the order of the output clocks ($\Phi_{M0\sim M3}$) of the main clock generator (Main CG), which generates multi-phase clocks based on a ring-counter, can always be guaranteed. The sub-clock generator (Sub CG) makes the input and common voltage sampling clocks ($\Phi_{S0,4,8,12}$ and $\Phi_{CM0,4,8,12}$) of the 4-channel sub-ADCs using the 5 GHz clock ($\Phi_{DL0}$). The delay matching (DM) logic is added to generate a delay corresponding to the gate delay of the sub-clock generator. Each sampling time skew of the THAs (i.e., the falling edge of $\Phi_{TH0\sim TH3}$) is controlled by the DCDL, and the digital input codes of the DCDL are obtained by the digital calibration engine implemented with the off-chip.

The details of the circuit implementations related to the clock generation will be covered in Section 3.3.

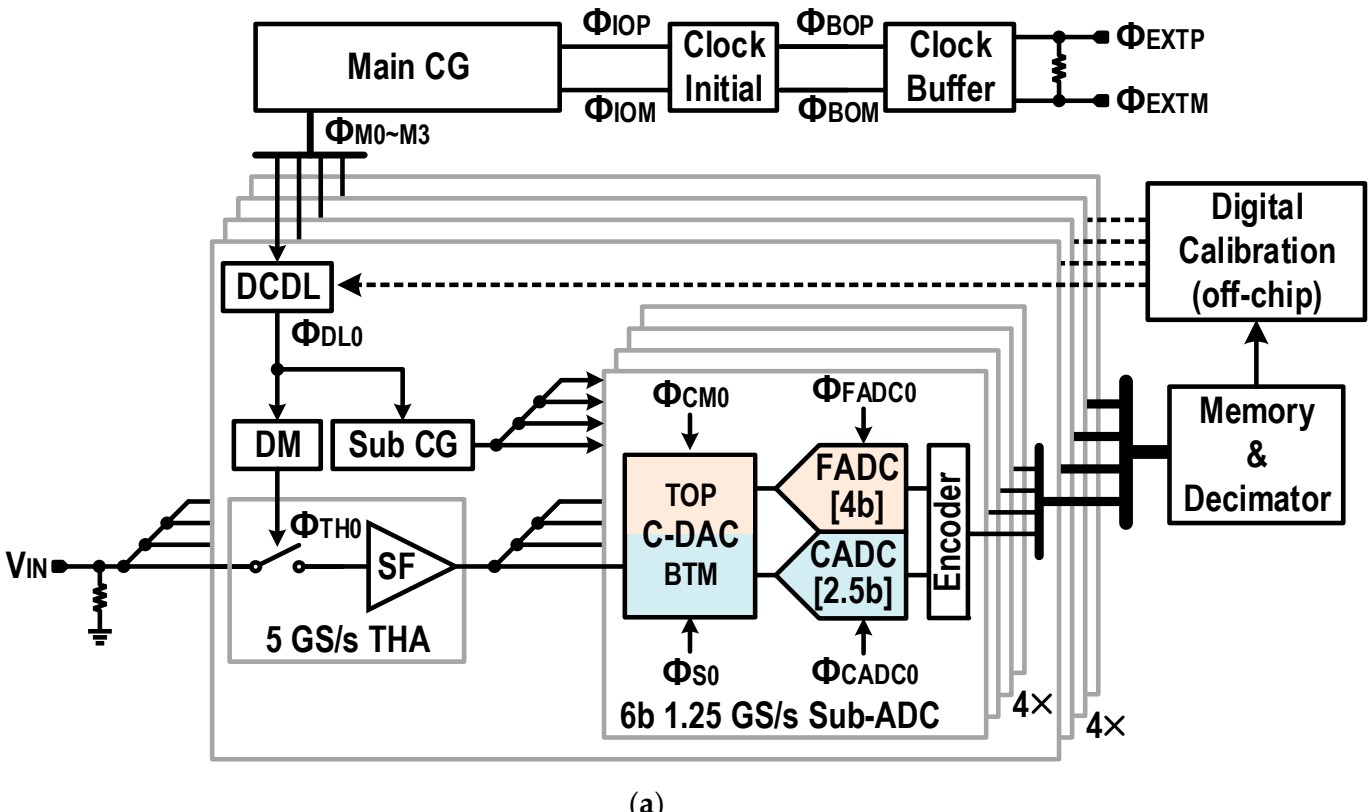

(**a**)

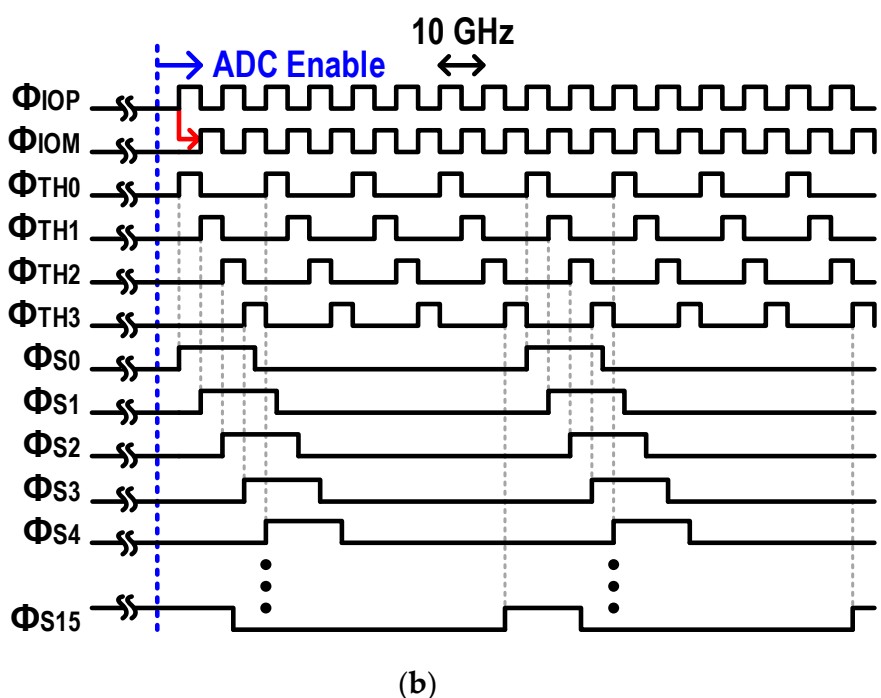

(**b**)

**Figure 1.** Proposed 6-bit 20 GS/s TI two-step flash ADC. (**a**) Block and (**b**) timing diagrams.

### 2.1. Input Network

Figure 2 shows the input sampling network of the proposed 20 GS/s TI ADC. In this design, 4-channel THAs are used to achieve a wide input bandwidth over the Nyquist input and to reduce sampling errors, such as a charge injection and clock feed-through, induced by the interference between the sub-ADCs. As shown in Figure 2, the THA consists of NMOS sampling switches ($M_1$ and $M_2$) with dummy switches ($M_3$–$M_6$) and PMOS-based SF buffers. The cross-coupled dummy switches ($M_3$ and $M_4$) prevent the signal feed-through during the hold time (i.e., $\Phi_{TH}$ is low), and the dummy switches ($M_5$ and $M_6$) with the source and drain nodes shorted compensate for the clock feed-through, which changes the input common voltage of the SF buffers [24,25]. Since the differential swing range of the input voltage and the input common mode voltage of the TI ADC are 400 mV and 200 mV, respectively, the clock boosting circuit for the NMOS sampling switches is not required. The tracking times for the track-and-hold (T/H) of the THA and the S/H of the sub-ADC are approximately 50 ps and 150 ps, respectively, at a sampling rate of 20 GS/s. At the falling edge of $\Phi_{TH}$, the input voltage is sampled to the parasitic capacitance ($C_P$), which is approximately 28 fF, formed by the SF buffer, dummy switches, and routing metal. Note that the sampling switches for the 4-channel THAs are not turned on at the same time. That is, the input bandwidth is defined by the resistance of the sampling switch and the parasitic capacitance for only one THA. Therefore, the input network of the proposed TI ADC is suitable for a wide input bandwidth [11]. Similar to the sampling method of the THA, the sampling switches of the 4-channel sub-ADCs driven by one SF buffer are not turned on at the same time. Moreover, due to the S/H sharing and reference-embedding techniques, the sampling capacitance of the sub-ADC was designed to be approximately 18 fF. As a result, the THA could be designed to be very compact, consuming 3.6 mW at a 0.9 V supply voltage.

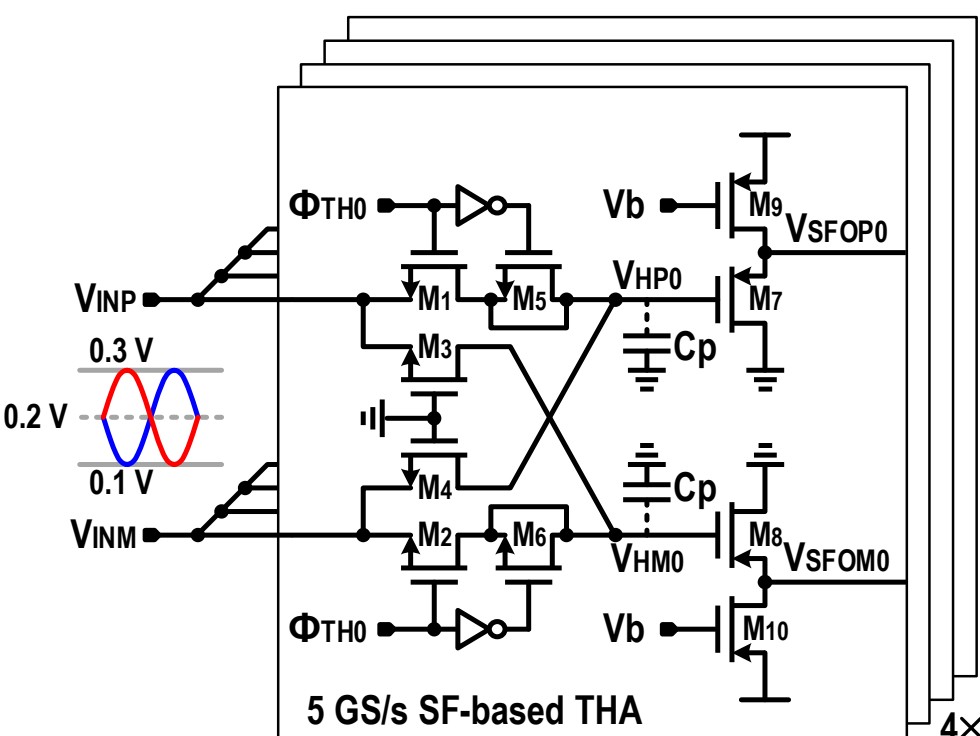

**Figure 2.** Input network of the proposed 20 GS/s TI ADC composed of 4-channel 5 GS/s SF-based THAs.

## 2.2. Two-Step Flash ADC

The small input capacitance, compact area, and low offset calibration burden of the sub-ADC are high priority factors for the realization of a power-efficient TI ADC with a wide input bandwidth. For this reason, in this design, the two-step flash architecture, considering the aforementioned factors, was applied to the sub-ADC for the TI ADC. The two-step flash ADC has been applied to the 7-bit 3 GS/s two-channel TI architecture in [22], and the advantages of the low-power, small area, and wide input bandwidth obtained by the S/H sharing and reference-embedding techniques have already been verified based on the measurement results. In this design, compared to [22], the two-step flash ADC was modified to be more suitable for the 20 GS/s TI ADC under the changed design conditions, such as the number of channels, input voltage range, and resolution.

The block diagram of the single-channel 6-bit 1.25 GS/s two-step flash ADC is shown in Figure 3. The ADC consists of a 2.5-bit CADC, a 4-bit reference-embedded 8-times interpolating FADC, a C-DAC for the input sampling and residue generation, a R-string for the references of both the CADC ($V_R$ [1:6]) and the C-DAC ($V_{RT}$ and $V_{RB}$), and a 6-bit digital encoder. Note that, unlike [22], a bootstrapped circuit for the input sampling switch is not used in this design. For a 6-bit resolution, the resolution of the CADC was selected to be 2.5-bit instead of 1.5-bit. The smaller resolution at the coarse stage can further reduce the input capacitance but consequently increases the hardware burden on the fine stage. Although the FADC can reduce its gate capacitance and power consumption thanks to the eight-time interpolation technique, the hardware complexity of the FADC increases relative to that of the CADC because the interpolation linearity and the offset calibration of the FADC should be considered. Moreover, the input capacitance of a single-channel ADC is mainly determined by the CDAC rather than the coarse comparator and is designed to be small enough thanks to the S/H sharing and the interpolation techniques.

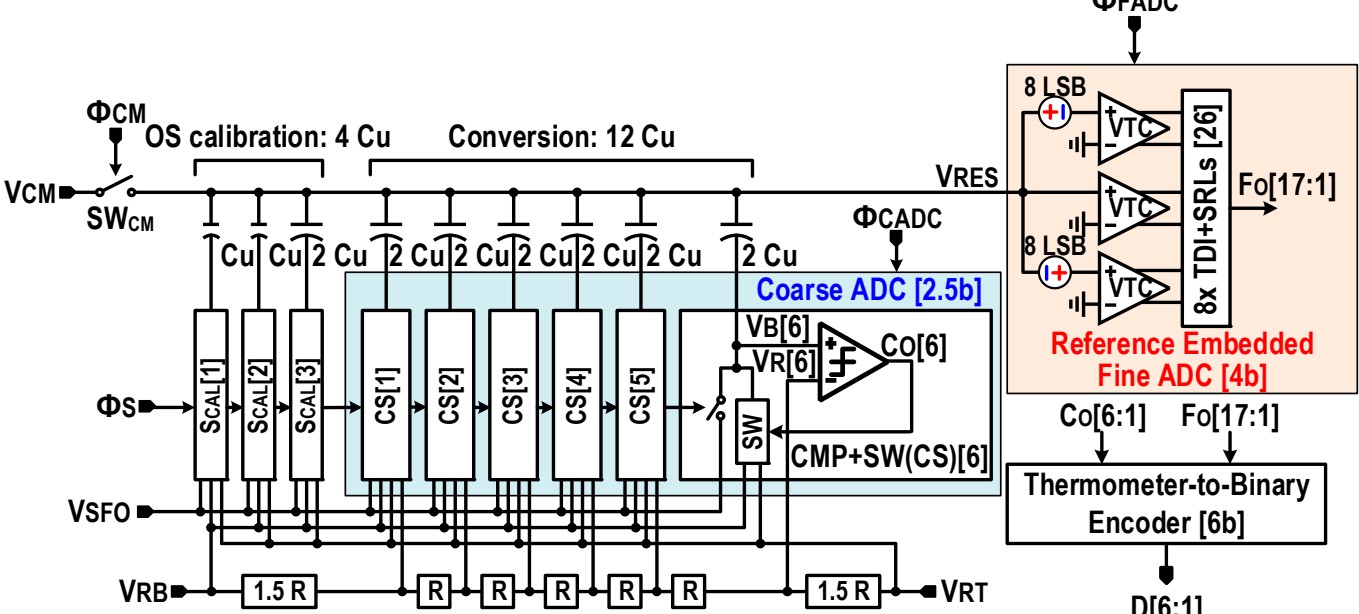

**Figure 3.** Block diagram of the 6-bit 1.25 GS/s two-step flash ADC.

In this paper, the operation of the two-step flash ADC is briefly explained. For details, please refer to my previous work [22]. The 2.5-bit CADC is a flash architecture consisting of six comparators and switches for the input sampling and residue generation. Each input node of the comparators is intentionally untied to use the six 2Cus as the C-DAC elements for the fine conversion. Note that, as the sampling is actually conducted in the top-plate sampling manner, there is no gain error between the sampled input and the reference voltages from the R-string regardless of the parasitic capacitance at the top-plate node.

The thermometer output codes of the comparators ($C_O$ [1:6]) directly control the switches and generate the residue voltage ($V_{RES}$) at the top-plate node of the C-DAC for the fine stage by using the six $2C_{US}$. The 4-bit FADC has only three reference-embedded VTCs at its front-end, and with the time-domain eight-time interpolation introduced in the flash ADC [26]. Thanks to the time-domain interpolation, the FADC greatly reduced the loading effect on the C-DAC; that is, the signal attenuation caused by the parasitic capacitance at the top-plate node of the C-DAC could be lowered. Savings in the power consumption and silicon area are other advantages of the interpolation technique. The reference-embedded VTC is covered in Section 3.2.

Looking at the timing diagram shown in Figure 4, the conversion clocks for the CADC and FADC ($\Phi_{CADC}$ and $\Phi_{FADC}$) are delayed versions of the input sampling clock ($\Phi_S$) and the common voltage sampling clock ($\Phi_{CM}$), respectively, which can be generated with a simple inverter delay; that is, the sub clock generator only needs to transmit $\Phi_S$ and $\Phi_{CM}$ to the sub-ADC. In this design, as the total capacitance of the C-DAC is as small as 12.8 fF (excluding parasitic) with $C_U$ = 0.8 fF, the settling time of the C-DAC is very short. This implies that the settling time can be generated by using a simple logic-gate delay from the falling edge of $\Phi_{CM}$. As a result, the remaining time after the fast C-DAC settling could be utilized for the FADC.

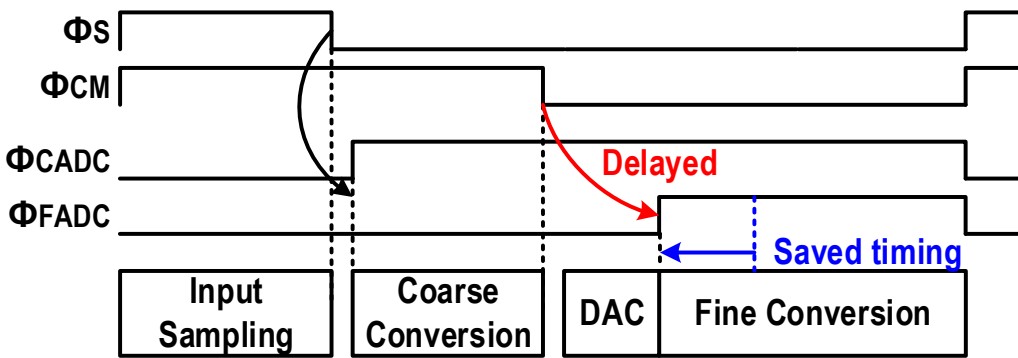

**Figure 4.** Timing diagram of the 6-bit 1.25 GS/s two-step flash ADC.

## 3. Circuit Implementation

### 3.1. Comparator Design for CADC

In this design, six dynamic comparators were applied to the 2.5-bit flash CADC. Since the unit capacitor of the C-DAC connected to the gate of the comparator is very small at 0.8 fF, it is necessary to consider the input kickback noise induced by the comparator. The kickback noise degrades the quality of the sampled signal to be used for the fine conversion in the proposed two-step ADC architecture. Therefore, as shown in Figure 5, the comparators were designed in a pseudo differential structure [22]. Note that the signal-dependent kickback error can be eliminated by keeping the source node of the input transistors to GND. Even though $V_{DP/DM}$ change during the latching operation, the kickback by them will be eliminated because the net voltages of $V_{DP/DM}$ are charged from GND and then discharged back to GND (eventually zero). Refer to [22] for more details on eliminating the kickback error. The estimated 1-sigma offset of the comparators is 15 mV, whereas the redundancy of the FADC is ±4 least significant bit (LSB), which is ±25 mV. Thus, the offset calibration of the comparators is required to prevent the nonlinearity caused by the offset mismatch. In this design, as shown in Figure 5, a differential pair connected in parallel with the input transistors was added for the offset calibration without a speed penalty. The gate voltages, $V_{CALP}$ and $V_{CALM}$, were controlled by a 4-bit resistive digital-to-analog converter (R-DAC) with a foreground calibration [26]. The range and accuracy of the offset calibration engine could be designed to be ±90 mV and 11.25 mV, respectively.

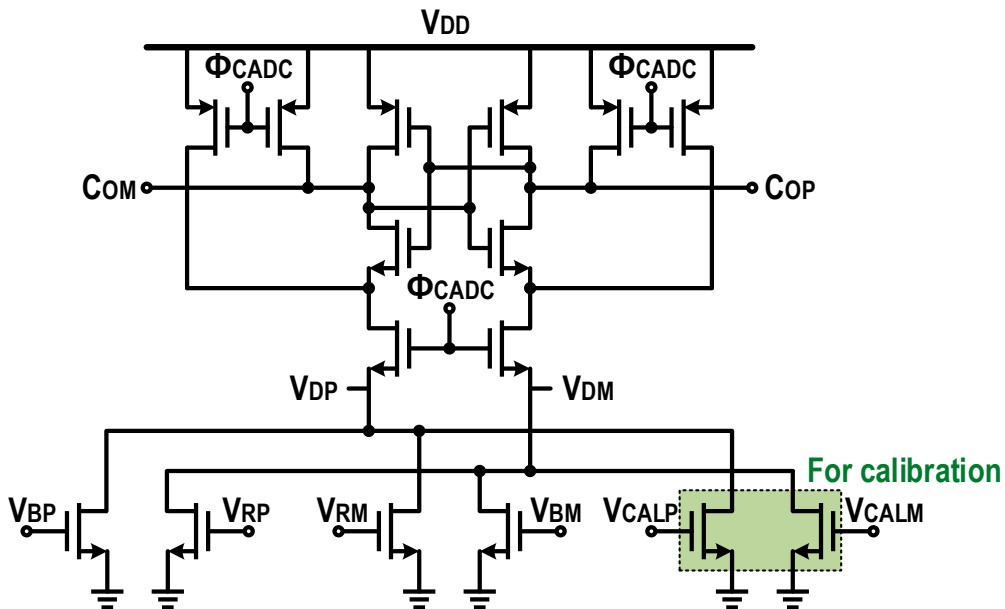

**Figure 5.** Pseudo differential comparator with the offset calibration capability used in the CADC.

### 3.2. Gain-Boosted VTC for FADC

Figure 6a shows the simplified block diagram of the 4-bit FADC with the eight-time time-domain interpolation technique. To generate 17 zero-crossing points (i.e., 4-bit resolution) in a flash-type conversion manner, the FADC consisted of three VTCs, a time-domain interpolator (TDI) array, and a NAND- and NOR-based SR-Latch array. The interpolation technique generates seven additional interpolated zero-crossing points between two adjacent VTCs. Thanks to the simple inverter-based TDIs, this structure reduces the power consumption, input capacitance, area, and burden of the offset calibration. The designed input capacitance of the FADC, including the routing capacitance between the C-DAC and the FADC, was only approximately 5 fF. This allowed the total capacitance of the C-DAC to be as low as 12.8 fF, resulting in an approximately 28% signal attenuation.

The circuit of the reference-embedded VTC is shown in Figure 6b. Note that, in this design, the 1 LSB voltage of the FADC was reduced by approximately 32% compared to [22]. Therefore, the voltage-to-time conversion gain must be increased to prevent a linearity degradation caused by the input-referred offset of the back-end circuits, such as the TDIs and SR-Latches. In this design, to enhance the time gain of the VTC, a positive feedback loop, which consists of M5 and M6, was added to the output nodes of the dynamic amplifier (i.e., $S_P$ and $S_M$ nodes) based on [27]. In addition, M7 and M8 was added to $S_P$ and $S_M$ nodes to alleviate the linearity degradation caused by the positive feedback. Consequently, the voltage-to-time gain of the VTC was increased from 0.9 ps/mV in [22] to 1.5 ps/mV.

The lower and upper VTCs (i.e., VTC<1> and VTC<3> shown in Figure 6a) have intentional offsets for embedded reference. Therefore, only one C-DAC with 16Cus was required for the residue generation, which means that the total number of the unit capacitance of the two-step flash ADC could be designed to be smaller than that of a conventional SAR ADC. The reference voltages for three VTCs were embedded by the different size ratio of M1 and M2 shown in Figure 6b. In this design, the size ratios of M1 and M2 were 4:2, 3:3, and 2:4 in that order for VTC<1:3> as shown in Figure 6a. The output time difference of the lower and upper VTCs (i.e., VTC<1> and VTC<3>) cannot be constant over a wide input range because of the transconductance (gm) nonlinearity of the input pair. However, in this design, the reference range between VTCs was approximately 32 mV, which is almost 2/3 of the previous work in [22], so the VTC gain could be designed to be more linear than that of [22].

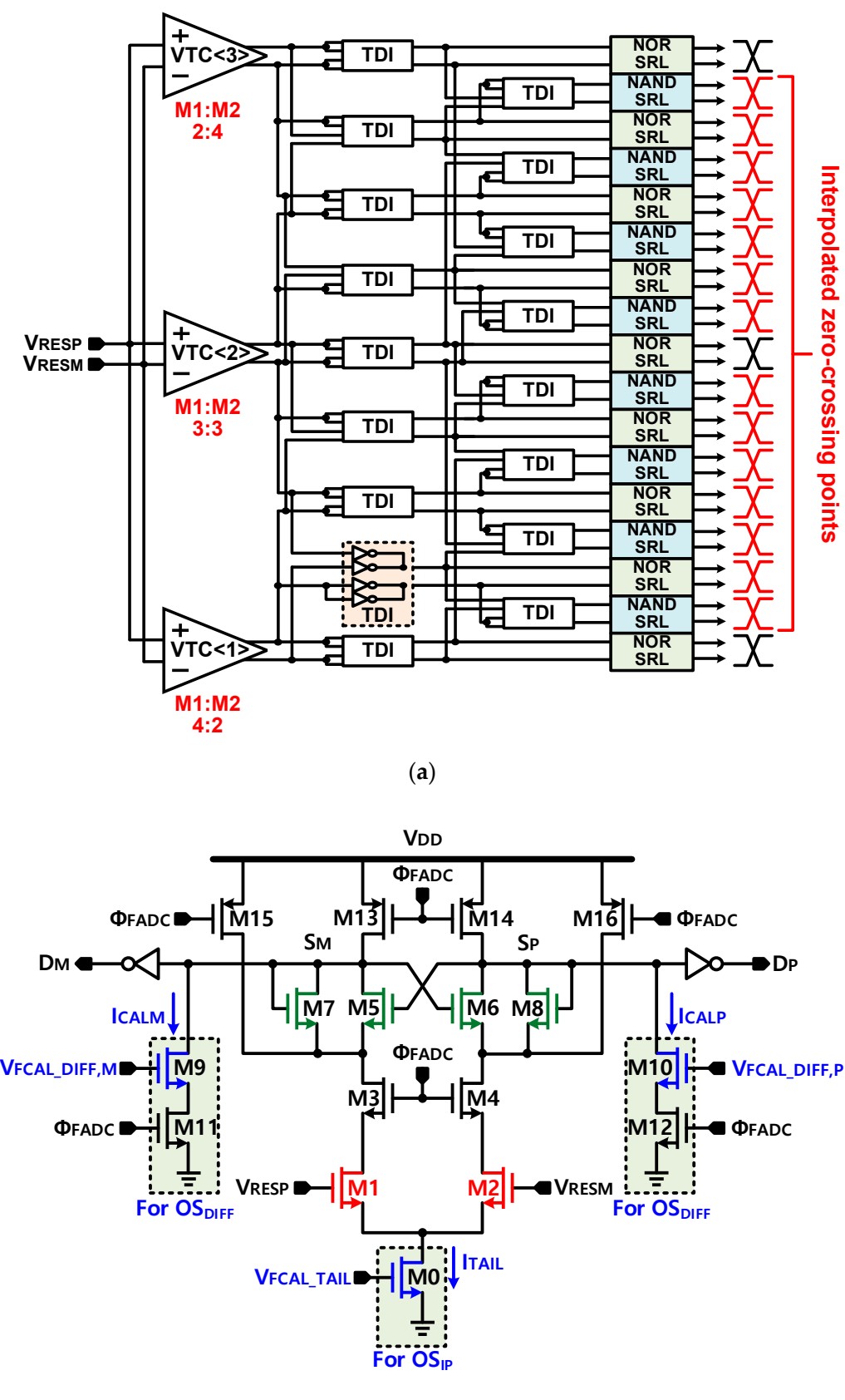

**Figure 6.** (**a**) Four-bit 8-time time-domain interpolating FADC. (**b**) Reference-embedded VTC with positive feedback for gain boosting.

On the other hand, the size ratio design cannot guarantee the required offset of each VTC accurately. Thus, an offset calibration for the VTCs is mandatory. In this design, the offset mismatches of the reference-embedded VTCs were calibrated by the advanced sequential slope-matching offset calibration technique in [22]. The embedded references of the three VTCs for the calibration were implanted during the foreground calibration mode by utilizing the 4Cus of the "OS calibration" part in the C-DAC shown in Figure 3, where the weight of Cu corresponds to the step size of the VTC. Note that, in this VTC-based time-domain interpolation, not only the differential offset of each VTC ($OS_{DIFF}$) but also the offsets of interpolated zero-crossing points generated by the outputs of two neighboring VTCs ($OS_{IP}$) should be taken care of with calibration. The $OS_{DIFF}$ and $OS_{IP}$ were calibrated by the differentially controlled current sources ($I_{CALP}$ and $I_{CALM}$) and by the tail current ($I_{TAIL}$), respectively, as shown in Figure 6b. One thing to note is that the change in the $I_{TAIL}$ can also change the $OS_{DIFF}$ as well because it affects the operating condition of the VTC. Therefore, even though the calibration for the $OS_{DIFF}$ is completed, it should be calibrated repeatedly until the calibration for the $OS_{IP}$ is carried out to compensate for the changes in the $OS_{DIFF}$. The design details of the advanced sequential slope-matching offset calibration were sufficiently covered in [22].

Figure 7 shows the output time difference according to the size ratios of the M5 (M6) and M7 (M8) of the VTC shown in Figure 6b. In this design, the size ratio of M5 and M7 was selected as 4:1 considering the linearity and the time gain of the VTC. As shown in Figure 7, when the size ratio of M5 and M7 is 4:1, the time gain varies from 1.56 ps/mV to 1.09 ps/mV within the interpolation range of the VTC; that is, the error of the time output due to the nonlinearity of the VTC is approximately 0.3 LSB, which is improved by half through the interpolation.

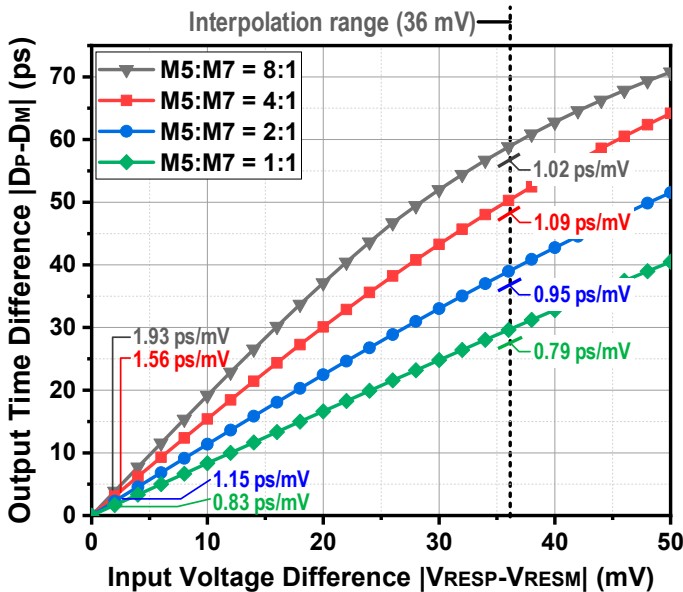

**Figure 7.** Output time difference of the VTC, $D_P$-$D_M$, versus various input voltages, $V_{RESP}$-$V_{RESM}$, according to the size ratios of M5 and M7 of the VTC.

### 3.3. High-Speed Multi-Phase Clock Generation

In order to implement a conversion rate of 20 GS/s in the proposed TI ADC, 5 GHz 4-phase sampling clocks ($\Phi_{M0\sim M3}$) and 1.25 GHz 16-phase sampling clocks ($\Phi_{S0\sim S15}$ and $\Phi_{CM0\sim CM15}$) are required for the 4-channel THAs and the 16-channel sub-ADCs, respectively. Since these high-speed multi-phase clocks have a very short time difference corresponding to 20 GHz (i.e., 50 ps), they need to be carefully designed to stably generate high-speed clocks. Therefore, this section deals with some design schemes for stably generating high-speed multi-phase clocks without systematic time mismatches between

the 4-channel THAs, including the DCDL logic for the monotonicity of the controlled sampling clocks.

### 3.3.1. Clock Initial Logic

In this design, to alleviate the design burden for the high-speed multi-phase clock generation, the proposed TI ADC received externally applied 10 GHz differential clocks ($\Phi_{EXTP}$ and $\Phi_{EXTM}$) instead of a 20 GHz single clock, and the differential clocks were transmitted to the input of the clock initial logic shown in Figure 8 through the clock buffer. Since the multi-phase clocks for the THAs are generated by dividing the two 10 GHz clocks ($\Phi_{IOP}$ and $\Phi_{IOM}$) based on a ring-counter, the order between $\Phi_{IOP}$ and $\Phi_{IOM}$ triggering the clock divider is very important. If the order between $\Phi_{IOP}$ and $\Phi_{IOM}$ is changed whenever the ADC is activated, the order of operation between the sub-channels is also changed, which can cause the TI ADC to malfunction. Therefore, the initial clock logic shown in Figure 8a was applied to the clock path so that the positive clock always precedes the negative clock among the two 10 GHz clocks. The operation of the clock initial logic can be described with Figure 8b. When $\Phi_{BOP}$ and $\Phi_{BOM}$ are free running, the reset signal (RST) of the ADC is converted from "High" to "Low" for starting the clock generation of the TI ADC. $EN_{Pre}$ and $EN_P$ are triggered to "High" by the first and second rising edges of $\Phi_{BOM}$, respectively, following the falling edge of RST. Keep in mind that the $DFF_1$ is added to alleviate the metastable issue of the D-type flip flop (DFF). $EN_M$ is triggered from "Low" to "High" only after $EN_P$ becomes "High" in synchronization with the following rising edge of $\Phi_{BOP}$. Therefore, the rising edge of $EN_M$ is always triggered with a delay of half of the 10 GHz clock period (i.e., 50 ps at 20 GS/s conversion rate) as that of $EN_P$. Finally, based on the triggered sequence of $EN_P$ and $EN_M$, $\Phi_{BOP}$ and $\Phi_{BOM}$ are bypassed by the two NAND gates to $\Phi_{IOP}$ and $\Phi_{IOM}$, respectively. One thing to note is that, for the normal clock generation, the sum of the gate delay and set-up time of the DFF should not exceed 50 ps at the 20 GS/s conversion rate. If the design margin is insufficient, an additional phase selector may be required. However, in this design, the sum of the gate delay and the set-up time of the DFF was designed to be approximately 42 ps in the slow condition.

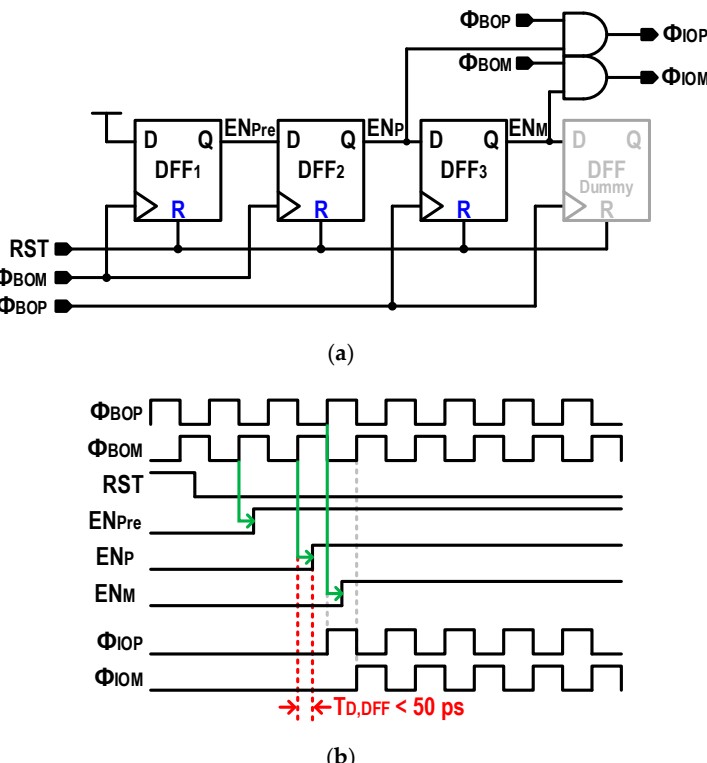

**Figure 8.** Clock initial logic. (**a**) Block and (**b**) timing diagrams.

### 3.3.2. Main Clock Generation

Figure 9a shows the block diagram of the main clock generator for the sampling clocks of the four-channel THAs. The main clock generator consists of two ring-counters, and has a DFF with a reset mode and a DFF with a set mode, and four NAND gates. As shown in Figure 9b, $\Phi_{IOP}$ and $\Phi_{IOM}$ drive their respective ring counter, and are divided into four-phase clocks, $\Phi_{DIV0\sim DIV3}$, with a 50% duty cycle, shifted by the phase difference between $\Phi_{IOP}$ and $\Phi_{IOM}$. Then, these four-phase clocks are converted to main sampling clocks, $\Phi_{M0\sim M3}$, with a 25% duty cycle for the THAs by the NAND gates. On the other hand, as shown in Figure 9c, the DFF is a true single-phase clock (TSPC) structure for low-power and high-speed clock triggering, and a dummy transistor was added to eliminate a phase mismatch caused by the transistors for the reset and set mode. Note that the phase mismatch in the path generating the sampling clock of the THAs is a factor that attenuates the accuracy of the time skew calibration engine. As mentioned in Section 3.3.1, since the multi-phase was generated using the 10 GHz differential clocks sequentially applied by the clock initial logic instead of a 20 GHz single clock, the design burden of the DFFs to make 5 GHz four-phase clocks was halved. As a result, designing a high-speed and power-efficient main clock generator for TI architecture corresponding to a 20 GHz clock speed could be simplified.

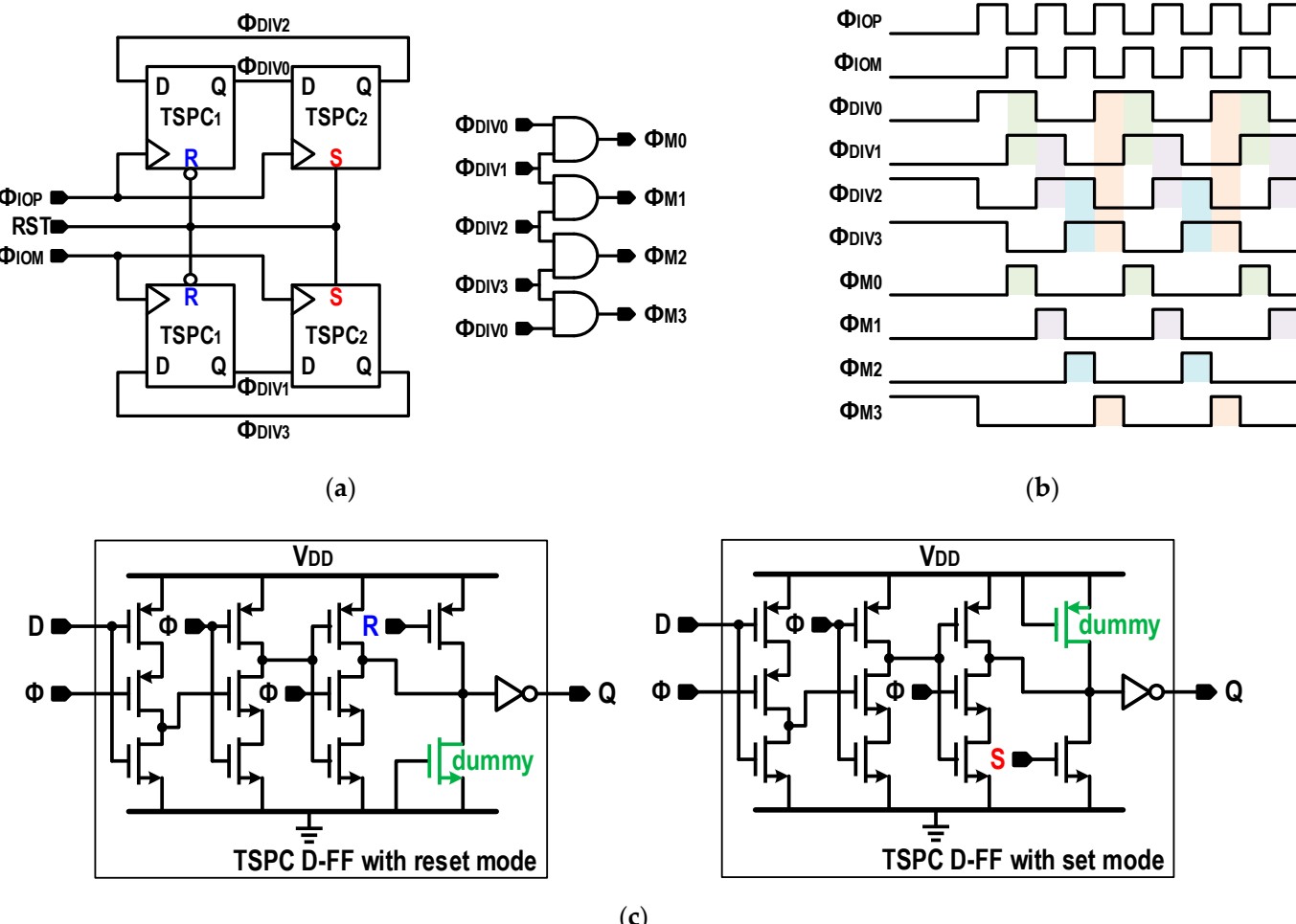

(**a**)                                                                 (**b**)

(**c**)

**Figure 9.** Main clock generator. (**a**) Multi-phase generation logic. (**b**) Timing diagram. (**c**) Circuits for the TSPC DFF with set and reset modes.

### 3.3.3. Digitally-Controlled Delay Line

The main sampling clocks ($\Phi_{TH0\sim TH3}$) of the four-channel THAs have a phase difference between the clocks due to process and layout mismatches. This phase difference causes the sampling time skew error between the THAs, which is a major source of the performance degradation according to increasing the input frequency in the TI architecture. Therefore, in this design, the time skew errors were calibrated on-chip by the DCDL shown in Figure 10, and detected off-chip based on [28,29]. Note that the time skew errors were detected based on a single-tone sinusoidal input. As shown in Figure 10, the 6-bit binary controlled DCDL consists of a capacitor array, an inverter chain, and a digital decoder. The capacitor array is divided into four banks, $DCDL_{S1\sim S4}$, in order to improve the calibration accuracy and relieve the driving strength of the inverter buffers. As shown in Table 1, each DCDL bank is sequentially selected from $DCDL_{S1}$ to $DCDL_{S4}$ by the thermometer control according to the 2-bit most significant bit (MSB) codes ($B_S$ [5:4]). In addition, the capacitor array of each DCDL bank is controlled by the 4-bit binary code ($B_{1\sim 4}$ [3:0]). Note that the $B_{1\sim 4}$ [3:0] is $B_S$ [3:0] bypassed to each DCDL bank according to $B_S$ [5:4].

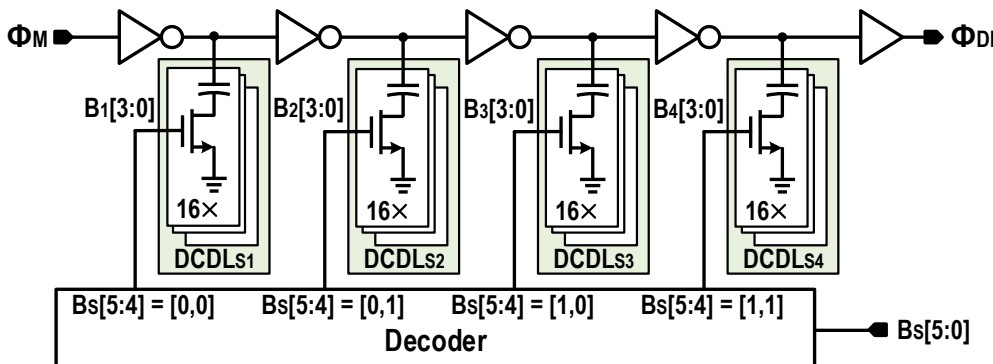

**Figure 10.** Block diagram of 6-bit DCDL.

**Table 1.** Truth table of DCDL.

| $B_S$ [5:4] | $B_1$ [3:0] | $B_2$ [3:0] | $B_3$ [3:0] | $B_4$ [3:0] |
|:---:|:---:|:---:|:---:|:---:|
| 00 | $B_S$ [3:0] | 0000 | 0000 | 0000 |
| 01 | 1111 | $B_S$ [3:0] | 0000 | 0000 |
| 10 | 1111 | 1111 | $B_S$ [3:0] | 0000 |
| 11 | 1111 | 1111 | 1111 | $B_S$ [3:0] |

### 3.3.4. Sub-Clock Generation

As mentioned in Section 2.2, only two 1.25 GHz clocks ($\Phi_S$ and $\Phi_{CM}$) are needed for the sub-ADC. The 1.25 GHz clocks can be generated by dividing the output of the DCDL, which is a 5 GHz clock, by four; that is, the clocks for four-channel sub-ADCs following one THA (e.g., $\Phi_{S0,4,8,12}$ and $\Phi_{CM0,4,8,12}$) can be generated with only one clock divider, such as a 4-bit ring counter shown in Figure 11a, which can lead to reducing the area and current consumption. In the process of generating $\Phi_M$ and $\Phi_S$, one thing to note is that the sampling clocks should be designed in consideration of an interference between the sub-ADCs. In other words, the falling edge of $\Phi_S$ should be non-overlapping with the rising edge of $\Phi_M$ for the other sub-ADC as shown in Figure 11b.

In this design, the non-overlap between the two clocks was realized using the DM logic shown in Figure 12. The DM logic guarantees that the rising edge of $\Phi_M$ always starts after the gate delay of two inverters rather than the falling edge of $\Phi_S$.

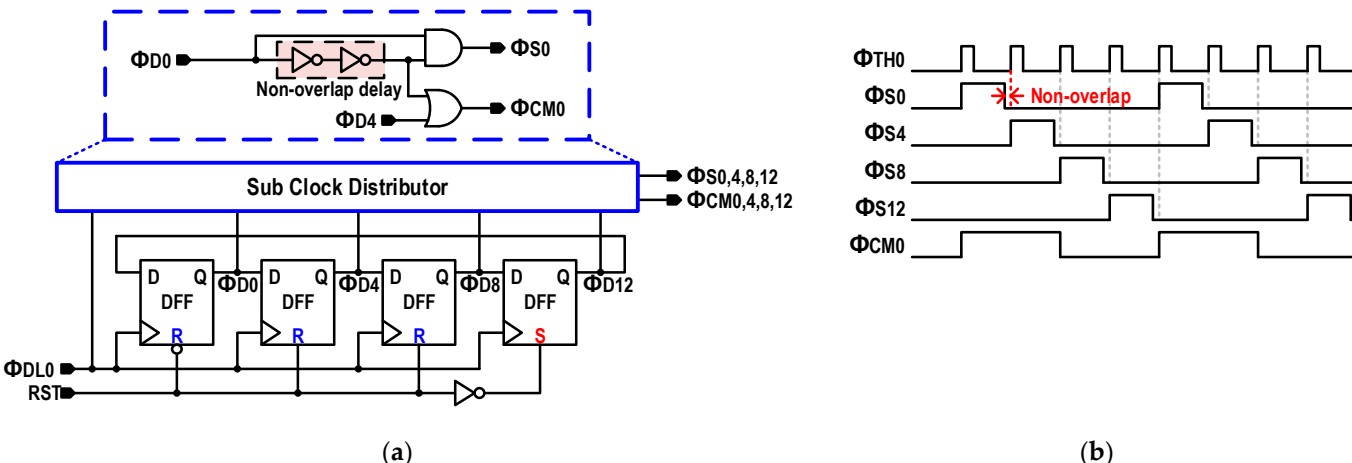

**Figure 11.** Sub clock generation for 4-channel sub-ADCs. (**a**) A 1.25 GHz 4-phase clock generator. (**b**) Timing diagram.

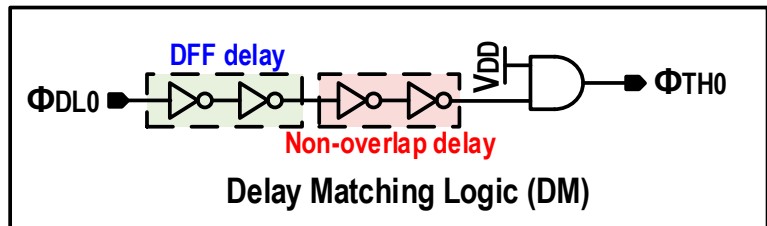

**Figure 12.** DM logic.

## 4. Measurement Results

A prototype 6-bit TI two-step flash ADC was designed to have a 20 GS/s conversion rate in a 40 nm CMOS process. Figure 13 shows a die photograph. An active area of the proposed TI ADC, including four-channel THAs, a main clock generator, four-channel sub-clock generators, and sixteen-channel sub-ADCs, is 0.1 mm$^2$, and the block for the foreground offset calibration occupies an additional area of 0.12 mm$^2$. The decimation logic and memory were implemented on-chip for real-time measurement and channel mismatch analysis, respectively. The memory can store 1024 samples (i.e., 64 samples per sub-ADC) for the 6-bit TI ADC.

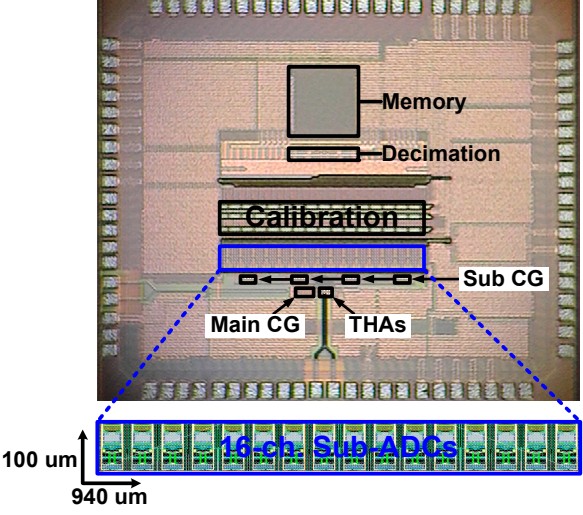

**Figure 13.** Die photograph.

Figure 14 shows the measurement setup for the 20 GS/s TI ADC. The input voltage and clock signals via balun and bias-tee were applied to inside the chip differentially matched to 100 ohm. Note that the differential signals, such as the input or clock, were differentially matched by the matched cables. The decimated and serialized digital output codes were transferred to the digital waveform analyzer to measure the performance in real-time and to detect the inter-channel mismatch errors, such as the offset, gain, and time skew.

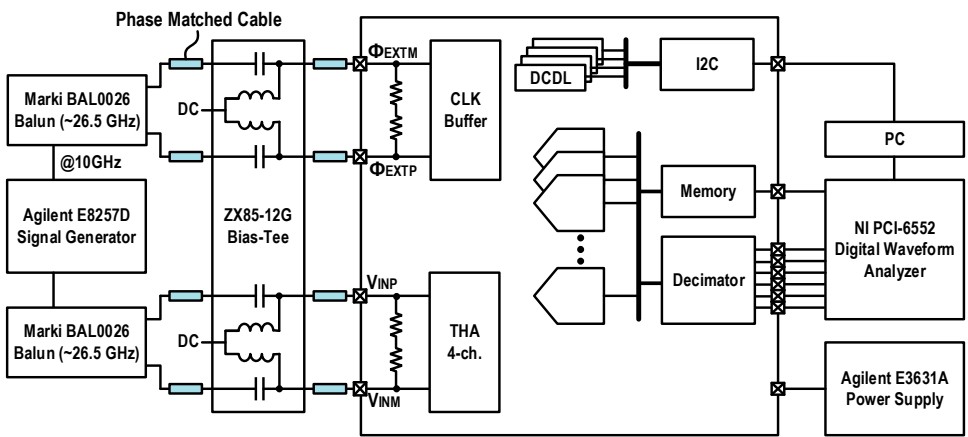

**Figure 14.** Measurement setup.

The measured differential non-linearity (DNL) and integral non-linearity (INL) profiles before and after the offset calibration for the coarse comparators and fine VTCs and the gain calibration for the four-channel THAs calibrated in the digital domain are shown in Figure 15. The peak DNL and INL improved from $+3.28/-0.64$ LSB and $+2.84/-2.84$ LSB to $+0.45/-0.31$ LSB and $+0.38/-0.38$ LSB, respectively.

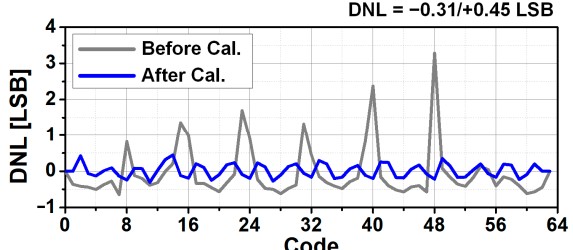
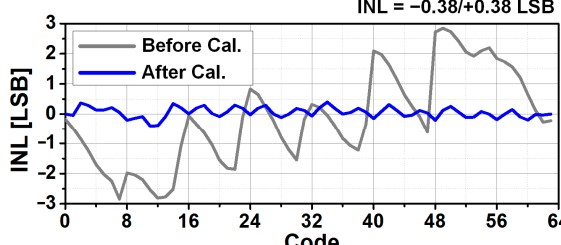

**Figure 15.** Measured DNL and INL.

Figure 16 shows the measured FFT spectra with a 0.828 GHz input at a sampling rate of 20 GS/s. The results are decimated by a factor of 459. The signal-to-noise ratio (SNR), spurious-free dynamic range (SFDR), and signal-to-noise and distortion ratio (SNDR) are improved from 14.46, 19.84, and 9.36 dB to 32.93, 42.75, and 32.58 dB, respectively, as shown in Figure 16a. The FFT spectrum without offset calibration shown in Figure 16a includes tones due to the offset mismatches between the interleaving channels as well as tones due to the offset mismatches of the comparators and VTCs used in the two-step flash ADC. Note that, since the offset mismatches for the comparators and VTCs were calibrated based on the same input common voltage used in the normal operation, the offset mismatches between the interleaving channels were also calibrated by the offset calibration process for the comparators and VTCs. Therefore, the noise floor as well as the interleaving offset tones could be improved by the offset calibration process for the comparators and VTCs. However, the gain mismatch tones between the interleaving channels still remain because of the gain mismatch between the THAs and the I-R drop of the reference voltage used by each C-DAC. These gain mismatch tones were calibrated in the digital domain based

on [28], and as a result, SNR, SFDR, and SNDR were improved to 34.02, 45.81, and 33.96 dB, respectively, as shown in Figure 16b.

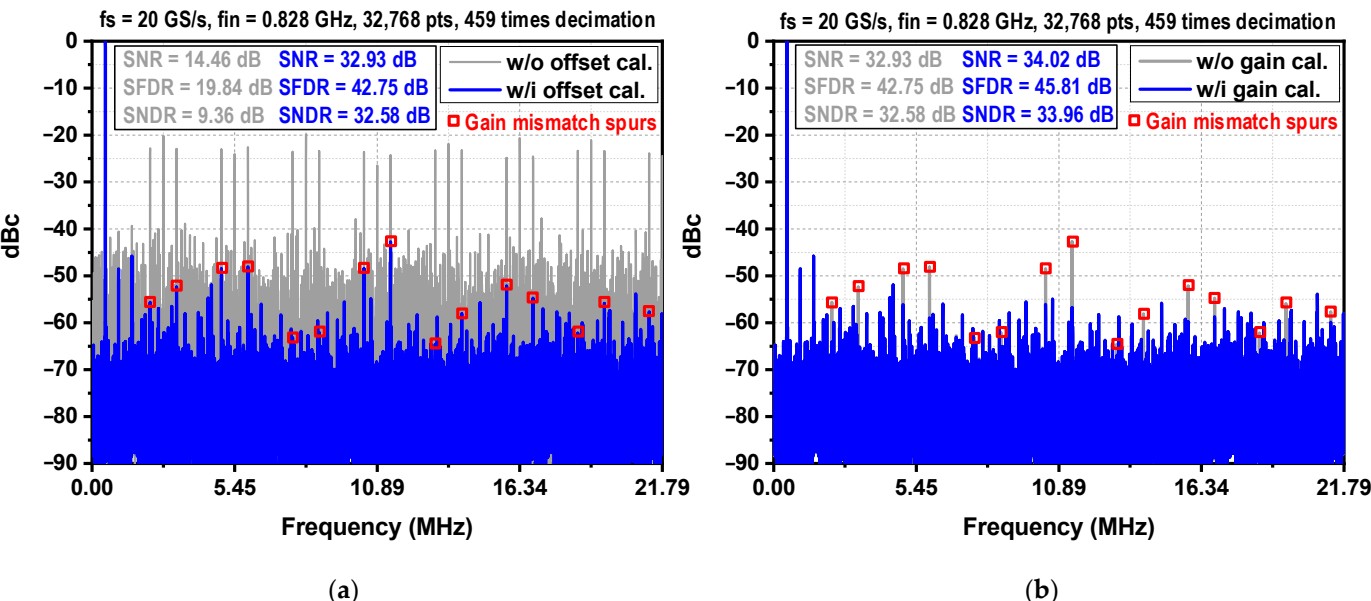

(**a**)                                                                                    (**b**)

**Figure 16.** Measured FFT spectra with a 0.828 GHz input at a 20 GS/s sampling rate. (**a**) With and without calibration for mismatches of comparators and VTCs. (**b**) With and without calibration for gain mismatch between the interleaving channels.

Figure 17 shows the FFT spectrum at a 9.042 GHz input. SNR, SFDR, and SNDR without skew calibration are 26.05, 30.62 and 25.69 dB, respectively, and limited by the timing skew error of approximately 950 fs, assuming only a timing skew error. As mentioned in Section 3.3.3, the timing skew error is detected off-chip, and then the sampling time is calibrated on-chip by the DCDL. The SNR, SFDR, and SNDR are improved to 31.02, 40.23, and 30.12 dB, respectively, through the skew calibration.

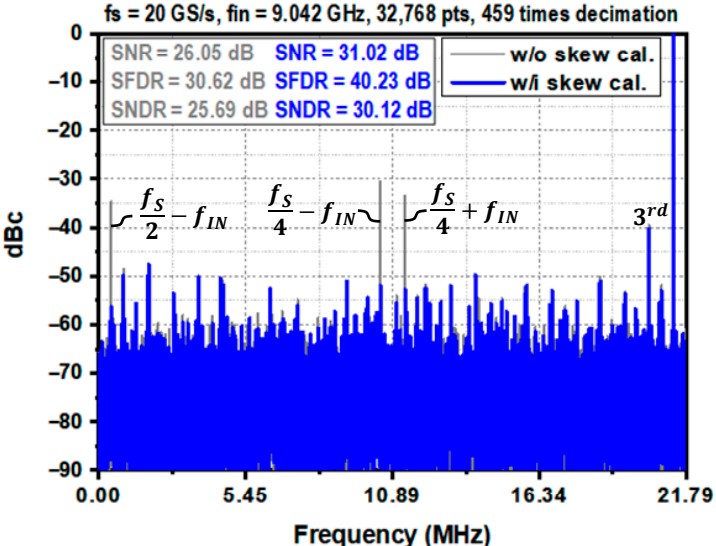

**Figure 17.** Measured FFT spectrum with and without skew calibration at 20 GS/s with a 9.085 GHz input.

Figure 18a shows the measured dynamic performances (SNDR and SFDR) at various input frequencies with the 20 GS/s sampling frequency. Thanks to the compact input

network due to the low input capacitance implemented by the proposed S/H sharing and the eight-time interpolation, the proposed TI ADC could be designed with a high effective resolution bandwidth (ERBW) of approximately 8.5 GHz when the time skew is optimally calibrated, as shown in Figure 18a. Without the skew calibration, SNDR begins to be attenuated by the skew error above an approximately 3 GHz input frequency. Figure 18b shows the measured SNDR and SFDR with various sampling frequencies at a 0.828 GHz input frequency. It can be seen that the proposed TI ADC operates stably up to a conversion rate of 20.3 GS/s, but the performance is rapidly degraded at conversion rates higher than that.

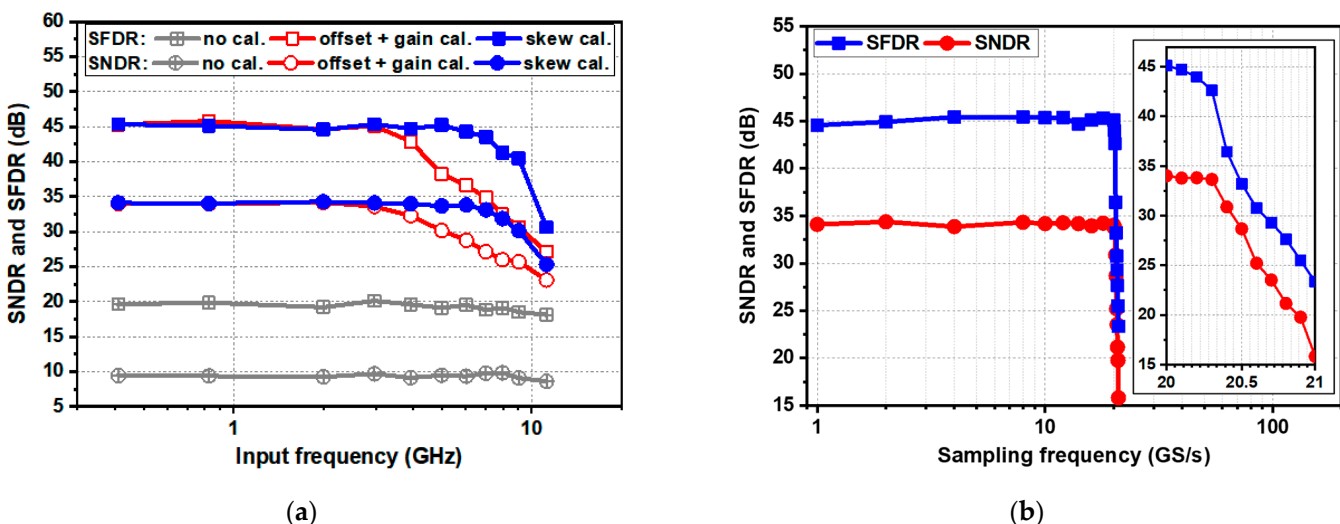

**(a)** **(b)**

**Figure 18.** Measured SNDR and SFDR versus various (**a**) input frequencies at 20 GS/s and (**b**) conversion rates with a 0.828 GHz input frequency.

The ADC core operates at a 0.9 V single supply and consumes 56.2 mW at a sampling rate of 20 GS/s. Figure 19 shows the power breakdown. The portion of the 16-channel sub-ADCs including the C-DAC with R-string is 32%, and the clock logics, including the main clock generator, sub clock generator, and the DCDL, take 32%. The digital encoder and calibration logic take 7.2% and 3.2%, respectively. Thanks to the small input capacitance of the sub-ADCs, each THA consumes 3.6 mW, so the four-channel THAs consume 14.4 mW, which is 25.6% of the total power consumption.

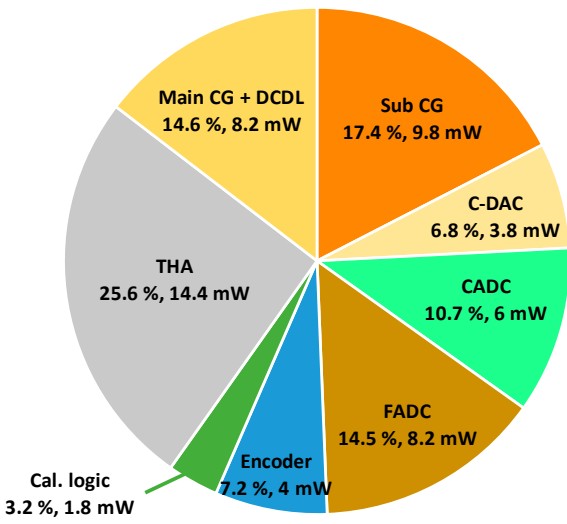

**Figure 19.** Power breakdown.

In Table 2, the performance of the proposed TI ADC is summarized and compared with those of recently reported 6-8-bit TI ADCs with sampling frequencies above 20 GS/s. The proposed 16-channel TI two-step flash ADC is capable of high-speed operation at 20 GS/s with a single supply voltage of 0.9 V. The proposed TI ADC could be designed with a competitive area and power consumption thanks to the S/H sharing technique and the reference-embedded eight-time interpolation technique using the gain enhanced VTC. As a result, the calculated Walden figure-of-merit (FoM) at the Nyquist input is 107.4 fJ/conversion step.

**Table 2.** Performance comparison.

| | This Work | JSSC14 V. Chen [8] | JSSC17 S. Cai [20] | VLSI16 Y. Frans [2] | JSSC17 B. Xu [30] | VLSI19 D. Pfaff [3] | SSCL20 S-J Kim [31] | VLSI21 M. Zhang [32] |
|---|---|---|---|---|---|---|---|---|
| Technology (nm) | 40 | 32 SOI | 65 | 16 FinFET | 28 | 7 FinFET | 16 FinFET | 65 |
| Architecture | TI Two-Step Flash | TI Flash | TI Multi-bit Search | TI SAR | TI SAR-TDC | TI SAR | TI Flash-TDC | TI Time-domain |
| # of channels | 16 | 8 | 8 | 32 | 16 | 32 | 16 | 8 |
| Supply (V) | 0.9 | 0.9 | 1.0 | 0.9/1.2/1.8 | 0.85/0.95 | - | 0.9 | 1.0/1.2 |
| Resolution (bit) | 6 | 6 | 6 | 8 | 6 | 8 | 8 | 8 |
| $F_S$ (GS/s) | 20 | 20 | 25 | 28 | 24 | 28 | 20 | 20 |
| $V_{IN}$ (mVdiff) | 400 | 300 | 500 | 1200 [2] | ~240 | - | 500 | 450 |
| $DNL/INL_{MAX}$ (LSB) | 0.45/0.38 | 0.47/0.42 | 0.64/0.60 | - | 0.25/0.22 | - | 0.95/2.39 | - |
| $SNDR_{@Nyq.}$ (dB) | 30.1 | 30.7 | 29.7 | 31.5 | 28.9 | 29.4 | 35.4 | 38.8 |
| $SFDR_{@Nyq.}$ (dB) | 40.2 | 39.4 | 42 | 39.1 | 41 | 39.1 | - | 52.5 |
| $ENOB_{@Nyq.}$ (bit) | 4.71 | 4.81 | 4.62 | 4.9 | 4.51 | 4.6 | 5.6 | 6.15 |
| Power (mW) | 56.2 | 69.5 | 88 | 280 | 23 | 150 | 175 | 129.9 |
| Active area ($mm^2$) | 0.1 | 0.25 | 0.24 | 2.8 [3] | 0.03 | 0.09 [4] | 0.1 | 0.22 |
| Walden FOM [1] (fJ/conv.-step) | 107.4 | 124.1 | 143 | 325.7 | 42 | 221 | 180 | 91.3 |

[1] Walden FoM = power/($2^{ENOB}$ × sampling frequency).  [2] Tx swing range.  [3] Dual transceiver active area.  [4] Including AFE.

## 5. Conclusions

This paper has presented a 40 nm CMOS 6-bit 20 GS/s 16-channel TI ADC using the two-step flash ADC with the S/H sharing and the reference-embedded eight-time interpolation techniques. The input bandwidth, area, and power efficiency of the single channel ADC could be improved by sharing a single S/H between the coarse and fine stages. The time gain of the VTC used in the eight-time time-domain interpolating FADC was boosted by the positive feedback loop, resulting in improving the linearity of the multi-bit interpolation. As a result of these, four-channel THAs consisting of the input network of the TI ADC could be implemented with a high speed and low power. In addition, the design burden of the multi-phase clock generation for the sub-ADCs could be alleviated by using the two-step flash structure. Thanks to these efforts, the prototype TI ADC operates at 20 GS/s with a single supply voltage as low as 0.9 V and consumes 56.2 mW. The effective number of bits (ENOB) at 0.828 GHz and 9.042 GHz inputs are 5.34 bit and 4.71 bit, respectively, and the Walden FoM at a 9.042 GHz input is 107.4 fJ/conversion step.

**Funding:** This research received no external funding.

**Conflicts of Interest:** The author declares no conflict of interest.

## Abbreviations

| | |
|---|---|
| ADC | Analog-to-Digital Converter |
| TI | Time-Interleaved |
| SAR | Successive Approximation Register |
| S/H | Sample-and-Hold |
| C-DAC | Capacitive Digital-to-Analog Converter |
| R-DAC | Resistive Digital-to-Analog Converter |
| R-string | Resistive string |
| THA | Track-and-Hold Amplifier |

| | |
|---|---|
| CADC | Coarse ADC |
| VTC | Voltage-to-Time Converter |
| FADC | Fine ADC |
| DFF | D-type Flip Flop |
| DCDL | Digitally-Controlled Delay Line |
| SF | Source Follower |
| LVDS | Low Voltage Differential Signaling |
| CG | Clock Generator |
| DM | Delay Matching |
| T/H | Track-and-Hold |
| MSB | Most Significant Bit |
| LSB | Least Significant Bit |
| TDI | Time-Domain Interpolator |
| TSPC | True Single Phase Clock |
| DNL | Differential Non-Linearity |
| INL | Integral Non-Linearity |
| SNR | Signal-to-Noise Ratio |
| SFDR | Spurious-Free Dynamic Range |
| SNDR | Signal-to-Noise and Distortion Ratio |
| ERBW | Effective Resolution Bandwidth |
| FoM | Figure of Merit |
| ENOB | Effective Number of Bits |

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
