# Peer review of "A 6-Bit 20 GS/s Time-Interleaved Two-Step Flash ADC in 40 nm CMOS"

_electronics, doi:10.3390/electronics11193052_

Round 1

Reviewer 1 Report

1Can the skew calibration module of this circuit calibrate the time skew error when inputting multi-frequency signals? How is the effect?

2How effective is the calibration capability of the pseudo-differential comparator in this paper? Please add specific simulation results?

3Please describe in detail how the gain-boosted VTC achieves a more linear gain? And supplement the corresponding formula derivation process?Please add relevant simulation results?

4There is not much description in the text about the shared sample and hold circuit in this design, so how does the author improve the input bandwidth, area and power efficiency of the single-channel ADC? Please add relevant details in detail?

Reviewer 2 Report

This work presented a 6-bit 20GS/s 16ch Time interleaved ADC based on a two-step flash ADC structure. With proposed S/H sharing and reference-embedding techniques, the bandwidth requirement can be met with minimized power consumption. Measured results are clearly presented to demonstrate the overall performance.

Although most of the techniques were proposed and covered in the author's previous work in Ref 22. This work added design consideration for a higher sampling speed, with modified S/H and VTC circuits. The overall work is solid and contains extensive consideration and discussion for the reader in the relevant field. Therefore, I recommend publication.

Author Response

I deeply appreciate your positive comments.

Reviewer 3 Report

The paper presents a high speed TI ADC for wireline applications. The paper is well structured and good details are provided.

Below are my comments:

- Please use Frequency rather than # of points as the x-axis in your FFT plot figures 15, 16, etc.

- Fig. 15a. It seems the SNR or the noise floor is worse w/o offset cal. Offset mismatches should cause tones at fixed locations, but here it seems the noise floor has gone up. Any possible explanation for this? Please comment.
